# Convex Potential Flows:
# Universal Probability Distributions with Optimal Transport and Convex Optimization

**Chin-Wei Huang**
University of Montreal & Mila
chin-wei.huang@umontreal.ca

**Ricky T. Q. Chen**
University of Toronto & Vector Institute
rtqichen@cs.toronto.edu

**Christos Tsirigotis**
University of Montreal & Mila
christos.tsirigotis@umontreal.ca

**Aaron Courville**
University of Montreal, Mila & CIFAR Fellow
aaron.courville@umontreal.ca

## Abstract

Flow-based models are powerful tools for designing probabilistic models with tractable density. This paper introduces Convex Potential Flows (CP-Flow), a natural and efficient parameterization of invertible models inspired by the optimal transport (OT) theory. CP-Flows are the gradient map of a strongly convex neural potential function. The convexity implies invertibility and allows us to resort to convex optimization to solve the convex conjugate for efficient inversion. To enable maximum likelihood training, we derive a new gradient estimator of the log-determinant of the Jacobian, which involves solving an inverse-Hessian vector product using the conjugate gradient method. The gradient estimator has *constant-memory* cost, and can be made effectively *unbiased* by reducing the error tolerance level of the convex optimization routine. Theoretically, we prove that CP-Flows are *universal* density approximators and are *optimal* in the OT sense. Our empirical results show that CP-Flow performs competitively on standard benchmarks of density estimation and variational inference.

## 1 Introduction

Normalizing flows (Dinh et al., 2014; Rezende & Mohamed, 2015) have recently gathered much interest within the machine learning community, ever since its recent breakthrough in modelling high dimensional image data (Dinh et al., 2017; Kingma & Dhariwal, 2018). They are characterized by an invertible mapping that can reshape the distribution of its input data into a simpler or more complex one. To enable efficient training, numerous tricks have been proposed to impose structural constraints on its parameterization, such that the density of the model can be tractably computed.

We ask the following question: "what is the natural way to parameterize a normalizing flow?" To gain a bit more intuition, we start from the one-dimension case. If a function $f : \mathbb{R} \to \mathbb{R}$ is continuous, it is invertible (injective onto its image) if and only if it is strictly monotonic. This means that if we are only allowed to move the probability mass continuously without flipping the order of the particles, then we can only rearrange them by changing the distance in between.

In this work, we seek to generalize the above intuition of monotone rearrangement in 1D. We do so by motivating the parameterization of normalizing flows from an optimal transport perspective, which allows us to define some notion of rearrangement cost (Villani, 2008). It turns out, if we want the output of a flow to follow some desired distribution, under mild regularity conditions, we can characterize the unique optimal mapping by a convex potential (Brenier, 1991). In light of this, we propose to parameterize normalizing flows by the gradient map of a (strongly) convex potential. Owing to this theoretical insight, the proposed method is provably *universal* and *optimal*; this means the proposed flow family can approximate arbitrary distributions and requires the least amount of transport cost. Furthermore, the parameterization with convex potentials allows us to formulate model inversion and gradient estimation as convex optimization problems. As such, we

make use of existing tools from the convex optimization literature to cheaply and efficiently estimate all quantities of interest.

In terms of the benefits of parameterizing a flow as a gradient field, the convex potential is an $\mathbb{R}^d \to \mathbb{R}$ function, which is different from most existing discrete-time flows which are $\mathbb{R}^d \to \mathbb{R}^d$. This makes CP-Flow relatively compact. It is also arguably easier to design a convex architecture, as we do not need to satisfy constraints such as orthogonality or Lipschitzness; the latter two usually require a direct or an iterative reparameterization of the parameters. Finally, it is possible to incorporate additional structure such as equivariance (Cohen & Welling, 2016; Zaheer et al., 2017) into the flow's parameterization, making CP-Flow a more flexible general purpose density model.

## 2 BACKGROUND: NORMALIZING FLOWS AND OPTIMAL TRANSPORT

Normalizing flows are characterized by a differentiable, invertible neural network $f$ such that the probability density of the network's output can be computed conveniently using the change-of-variable formula

$$p_Y(f(x)) = p_X(x)\left|\frac{\partial f(x)}{\partial x}\right|^{-1} \qquad \Longleftrightarrow \qquad p_Y(y) = p_X(f^{-1}(y))\left|\frac{\partial f^{-1}(y)}{\partial y}\right| \qquad (1)$$

where the Jacobian determinant term captures the local expansion or contraction of the density near $x$ (resp. $y$) induced by the mapping $f$ (resp. $f^{-1}$), and $p_X$ is the density of a random variable $X$. The invertibility requirement has led to the design of many special neural network parameterizations such as triangular maps, ordinary differential equations, orthogonality or Lipschitz constraints.

**Universal Flows** For a general learning framework to be meaningful, a model needs to be flexible enough to capture variations in the data distribution. In the context of density modeling, this corresponds to the model's capability to represent arbitrary probability distributions of interest. Even though there exists a long history of literature on universal approximation capability of deep neural networks (Cybenko, 1989; Lu et al., 2017; Lin & Jegelka, 2018), invertible neural networks generally have limited expressivity and cannot approximate arbitrary functions. However, for the purpose of approximating a probability distribution, it suffices to show that the distribution induced by a normalizing flow is universal.

Among many ways to establish distributional universality of flow based methods (*e.g.* Huang et al. 2018; 2020b; Teshima et al. 2020; Kong & Chaudhuri 2020), one particular approach is to approximate a *deterministic coupling* between probability measures. Given a pair of probability densities $p_X$ and $p_Y$, a deterministic coupling is a mapping $g$ such that $g(X) \sim p_Y$ if $X \sim p_X$. We seek to find a coupling that is invertible, or at least can be approximated by invertible mappings.

**Optimal Transport** Let $c(x, y)$ be a cost function. The *Monge problem* (Villani, 2008) pertains to finding the optimal transport map $g$ that realizes the minimal expected cost

$$J_c(p_X, p_Y) = \inf_{\widetilde{g}:\widetilde{g}(X)\sim p_Y} \mathbb{E}_{X\sim p_X}[c(X, \widetilde{g}(X))] \qquad (2)$$

When the second moments of $X$ and $Y$ are both finite, and $X$ is regular enough (e.g. having a density), then the special case of $c(x, y) = ||x - y||^2$ has an interesting solution, a celebrated theorem due to Brenier (1987; 1991):

**Theorem 1** (**Brenier's Theorem**, Theorem 1.22 of Santambrogio (2015)). *Let $\mu, \nu$ be probability measures with a finite second moment, and assume $\mu$ has a Lebesgue density $p_X$. Then there exists a convex potential $G$ such that the gradient map $g := \nabla G$ (defined up to a null set) uniquely solves the Monge problem in eq. (2) with the quadratic cost function $c(x, y) = ||x - y||^2$.*

Some recent works are also inspired by Brenier's theorem and utilize a convex potential to parameterize a critic model, starting from Taghvaei & Jalali (2019), and further built upon by Makkuva et al. (2019) who parameterize a generator with a convex potential and concurrently by Korotin et al. (2019). Our work sets itself apart from these prior works in that it is entirely likelihood-based, minimizing the (empirical) KL divergence as opposed to an approximate optimal transport cost.

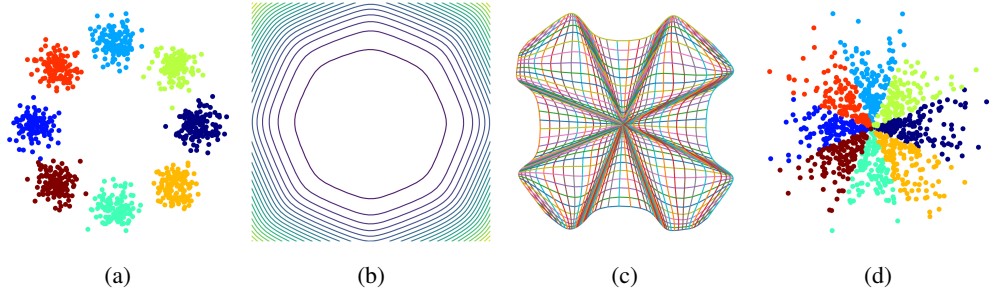

|        |        |        |        |
|:------:|:------:|:------:|:------:|
| (a)    | (b)    | (c)    | (d)    |

Figure 1: Illustration of Convex Potential Flow. (a) Data $x$ drawn from a mixture of Gaussians. (b) Learned convex potential $F$. (c) Mesh grid distorted by the gradient map of the convex potential $f = \nabla F$. (d) Encoding of the data via the gradient map $z = f(x)$. Notably, the encoding is the *value of the gradient* of the convex potential. When the curvature of the potential function is locally flat, gradient values are small and this results in a contraction towards the origin.

## 3   CONVEX POTENTIAL FLOWS

Given a strictly convex potential $F$, we can define an injective map (invertible from its image) via its gradient $f = \nabla F$, since the Jacobian of $f$ is the Hessian matrix of $F$, and is thus positive definite. In this section, we discuss the parameterization of the convex potential $F$ (3.1), and then address gradient estimation for CP-Flows (3.2). We examine the connection to other parameterization of normalizing flows (3.3), and finally rigorously prove universality in the next section.

### 3.1   MODELING

**Input Convex Neural Networks**   We use $L(x)$ to denote a linear layer, and $L^+(x)$ to denote a linear layer with positive weights. We use the (fully) input-convex neural network (ICNN, Amos et al. (2017)) to parameterize the convex potential, which has the following form

$$F(x) = L^+_{K+1}(s(z_K)) + L_{K+1}(x) \qquad z_k := L^+_k(s(z_{k-1})) + L_k(x) \qquad z_1 := L_1(x)$$

where $s$ is a non-decreasing, convex activation function. In this work, we use softplus-type activation functions, which is a rich family of activation functions that can be shown to uniformly approximate the ReLU activation. See Appendix B for details.

**Invertibility and Inversion Procedure**   If the activation $s$ is twice differentiable, then the Hessian $H_F$ is positive semi-definite. We can make it strongly convex by adding a quadratic term $F_\alpha(x) = \frac{\alpha}{2}\|x\|_2^2 + F(x)$, such that $H_{F_\alpha} \succeq \alpha I \succ 0$. This means the gradient map $f_\alpha = \nabla F_\alpha$ is injective onto its image. Furthermore, it is surjective since for any $y \in \mathbb{R}^d$, the potential $x \mapsto F_\alpha(x) - y^\top x$ has a unique minimizer[1] satisfying the first order condition $\nabla F_\alpha(x) = y$, due to the strong convexity and

---

**Algorithm 1** Inverting CP-Flow.

---

1: **procedure** INVERT($F, y, \texttt{CvxSolver}$)
2:     Initialize $x \leftarrow y$
3:     **def** closure():
4:         Compute loss: $l \leftarrow F(x) - y^\top x$
5:         **return** $l$
6:     $x \leftarrow \texttt{CvxSolver}(\texttt{closure}, x)$
7:     **return** $x$

---

differentiability. We refer to this invertible mapping $f_\alpha$ as the *convex potential flow*, or the CP-Flow. The above discussion also implies we can plug in a black-box convex solver to invert the gradient map $f_\alpha$, which we summarize in Algorithm 1. Inverting a batch of independent inputs is as simple as summing the convex potential over all inputs: since all of the entries of the scalar $l$ in the minibatch are independent of each other, computing the gradient all $l$'s wrt all $x$'s amounts to computing the gradient of the summation of $l$'s wrt all $x$'s. Due to the convex nature of the problem, a wide selection of algorithms can be used with convergence guarantees (Nesterov, 1998). In practice, we use the *L-BFGS* algorithm (Byrd et al., 1995) as our CvxSolver.

---

[1] The minimizer $x^*$ corresponds to the gradient map of the *convex conjugate* of the potential. See Appendix A for a formal discussion.

**Estimating Log Probability** Following equation (1), computing the log density for CP-Flows requires taking the log determinant of a symmetric positive definite Jacobian matrix (as it is the Hessian of the potential). There exists numerous works on estimating spectral densities (*e.g.* Tal-Ezer & Kosloff, 1984; Silver & Röder, 1994; Han et al., 2018; Adams et al., 2018), of which this quantity is a special case. See Lin et al. (2016) for an overview of methods that only require access to Hessian-vector products. Hessian-vector products (hvp) are cheap to compute with reverse-mode automatic differentiation (Baydin et al., 2017), which does not require constructing the full Hessian matrix and has the same asymptotic cost as evaluating $F_\alpha$.

In particular, the log determinant can be rewritten in the form of a generalized trace $\mathrm{tr}\log H$. Chen et al. (2019a) limit the spectral norm (*i.e.* eigenvalues) of $H$ and directly use the Taylor expansion of the matrix logarithm. Since our $H$ has unbounded eigenvalues, we use a more complex algorithm designed for symmetric matrices, the *stochastic Lanczos quadrature* (SLQ; Ubaru et al., 2017). At the core of SLQ is the Lanczos method, which computes $m$ eigenvalues of $H$ by first constructing a symmetric tridiagonal matrix $T \in \mathbb{R}^{m \times m}$ and computing the eigenvalues of $T$. The Lanczos procedure only requires Hessian-vector products, and it can be combined with a stochastic trace estimator to provide a stochastic estimate of our log probability. We chose SLQ because it has shown theoretically and empirically to have low variance (Ubaru et al., 2017).

## 3.2 $\mathcal{O}(1)$-Memory Unbiased $\nabla \log \det H$ estimator

We would also like to have an estimator for the *gradient* of the log determinant to enable variants of stochastic gradient descent for optimization. Unfortunately, directly backpropagating through the log determinant estimator is not ideal. Two major drawbacks of directly differentiating through SLQ are that it requires (i) differentiating through an eigendecomposition routine and (ii) storing all Hessian-vector products in memory (see fig. 2). Problem (i) is more specific to SLQ, because the gradient of an eigendecomposition is not defined when the eigenvalues

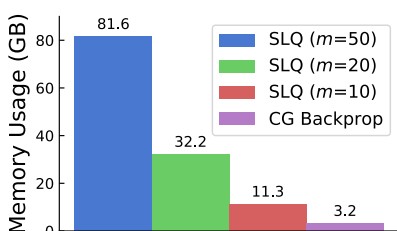

Figure 2: Memory for training CIFAR-10.

are not unique (Seeger et al., 2017). Consequently, we have empirically observed that differentiating through SLQ can be unstable, frequently resulting in NaNs due to the eigendecomposition. Problem (ii) will hold true for other algorithms that also estimate $\log \det H$ with Hessian-vector products, and generally the only difference is that a different numerical routine would need to be differentiated through. Due to these problems, we do not differentiate through SLQ, but we still use it as an efficient method for monitoring training progress.

Instead, it is possible to construct an alternative formulation of the gradient as the solution of a convex optimization problem, foregoing the necessity of differentiating through an estimation routine of the log determinant. We adapt the gradient formula from Chen et al. (2019a, Appendix C) to the context of convex potentials. Using Jacobi's formula[*] and the adjugate representation of the matrix inverse[†], for any invertible matrix $H$ with parameter $\theta$, we have the following identity:

$$\frac{\partial}{\partial \theta} \log \det H = \frac{1}{\det H} \frac{\partial}{\partial \theta} \det H \stackrel{*}{=} \frac{1}{\det H} \mathrm{tr}\left(\mathrm{adj}(H)\frac{\partial H}{\partial \theta}\right) \stackrel{\dagger}{=} \mathrm{tr}\left(H^{-1}\frac{\partial H}{\partial \theta}\right) = \mathbb{E}_v\left[v^\top H^{-1}\frac{\partial H}{\partial \theta}v\right]. \tag{3}$$

Notably, in the last equality, we used the Hutchinson trace estimator (Hutchinson, 1989) with a Rademacher random vector $v$, leading to a $\mathcal{O}(1)$-memory, unbiased Monte Carlo gradient estimator.

Computing the quantity $v^\top H^{-1}$ in eq. (3) by constructing and inverting the full Hessian requires $d$ calls to an automatic differentiation routine and is too costly for our purposes. However, we can recast this quantity as the solution of a quadratic optimization problem

$$\arg\min_z \left\{ \frac{1}{2}z^\top H z - v^\top z \right\} \tag{4}$$

which has the unique minimizer $z^* = H^{-1}v$ since $H$ is symmetric positive definite.

We use the *conjugate gradient* (CG) method, which is specifically designed for solving the unconstrained optimization problems in eq. (4) with symmetric positive definite $H$. It uses only Hessian-vector products and is straightforward to parallelize. Conjugate gradient is guaranteed to return the exact solution $z^*$ within $d$ iterations, and the error of the approximation is known to converge exponentially fast $||z^m - z^*||_H \leq 2\gamma^m ||z^0 - z^*||_H$, where $z^m$ is the estimate after

---

**Algorithm 2** Surrogate training objective.

1: **procedure** SURROGATEOBJ($F, x,$ CG)
2:     Obtain the gradient $f(x) \triangleq \nabla_x F(x)$
3:     Sample Rademacher random vector $r$
4:     **def** hvp($v$):
5:         **return** $v^\top \frac{\partial}{\partial x} f(x)$
6:     $z \leftarrow$ stop_gradient (CG(hvp, $r$))
7:     **return** hvp($z$)$^\top r$

---

$m$ iterations. The rate of convergence $\gamma < 1$ relates to the condition number of $H$. For more details, see Nocedal & Wright (2006, Ch. 5). In practice, we terminate CG when $||Hz^m - v||_\infty < \tau$ is satisfied for some user-controlled tolerance. Empirically, we find that stringent tolerance values are unnecessary for stochastic optimization (see appendix F).

Estimating the full quantity in eq. (3) is then simply a matter of computing and differentiating a scalar quantity (a surrogate objective) involving another Hessian-vector product: $\frac{d}{d\theta}\left((z^m)^\top Hv\right)$, where only $H$ is differentiated through (since $z^m$ is only used to approximate $v^\top H^{-1}$ as a modifier of the gradient). We summarize this procedure in Algorithm 2. Similar to inversion, the hvp can also be computed in batch by summing over the data index, since all entries are independent.

### 3.3 CONNECTION TO OTHER NORMALIZING FLOWS

**Residual Flow** For $\alpha = 1$, the gradient map $f_1$ resembles the residual flow (Behrmann et al., 2019; Chen et al., 2019a). They require the residual block—equivalent to our gradient map $f$—to be *contractive* (with Lipschitz constant strictly smaller than 1) as a sufficient condition for invertibility. In contrast, we enforce invertibility by using strongly convex potentials, which guarantees that the inverse of our flow is globally unique. With this, we do not pay the extra compute cost for having to satisfy Lipschitz constraints using methods such as spectral normalization (Miyato et al., 2018). Our gradient estimator is also derived similarly to that of Chen et al. (2019a), though we have the benefit of using well-studied convex optimization algorithms for computing the gradients.

**Sylvester Flow** By restricting the architecture of our ICNN to one hidden layer, we can also recover a form similar to Sylvester Flows. For a 1-hidden layer ICNN ($K = 1$) and $\alpha = 1$, we have $F_1 = \frac{1}{2}||x||_2^2 + L_2^+(s(L_1x)) + L_2(x)$. Setting the weights of $L_2$ to zero, we have

$$f_1(x) = \nabla_x F_1(x) = x + W_1^\top \text{diag}(w_2^+)s'(W_1x + b_1). \tag{5}$$

We notice the above form bears a close resemblance to the Sylvester normalizing flow (Van Den Berg et al., 2018) (with $\boldsymbol{Q}$, $\boldsymbol{R}$ and $\widetilde{\boldsymbol{R}}$ from Van Den Berg et al. (2018) being equal to $W_1^\top$, $\text{diag}(w_2^+)$ and $I$, respectively). For the Sylvester flow to be invertible, they require that $\boldsymbol{R}$ and $\widetilde{\boldsymbol{R}}$ be triangular and $\boldsymbol{Q}$ be orthogonal, which is a computationally costly procedure. This orthogonality constraint also implies that the number of hidden units cannot exceed $d$. This restriction to orthogonal matrices and one hidden layer are for applying Sylvester's determinant identity. In contrast, we do not require our weight matrices to be orthogonal, and we can use any hidden width and depth for the ICNN.

**Sigmoidal Flow** Let $s$ be the softplus activation function and $\sigma = s'$. Then for the 1-dimensional case ($d = 1$) and $\alpha = 0$ (without the residual connection), we have

$$\frac{\partial}{\partial x}F_0(x) = \sum_{j=1} w_{1,j}w_{2,j}^+ \sigma(w_{1,j}x + b_{1,j}) = \sum_{j=1} |w_{1,j}|w_{2,j}^+ \sigma(|w_{1,j}|x + \text{sign}(w_{1,j})b_{1,j}) + \text{const}. \tag{6}$$

which is equivalent to the sigmoidal flow of Huang et al. (2018) up to rescaling (since the weighted sum is no longer a convex sum) and a constant shift, and is monotone due to the positive weights. This correspondence is not surprising since a differentiable function is convex if and only if its derivative is monotonically non-decreasing. It also means we can parameterize an increasing function as the derivative of a convex function, which opens up a new direction for parameterizing autoregressive normalizing flows (Kingma et al., 2016; Huang et al., 2018; Müller et al., 2019; Jaini et al., 2019; Durkan et al., 2019; Wehenkel & Louppe, 2019).

**Flows with Potential Parameterization**    Inspired by connections between optimal transport and continuous normalizing flows, some works (Zhang et al., 2018; Finlay et al., 2020a; Onken et al., 2020) have proposed to parameterize continuous-time transformations by taking the gradient of a scalar potential. They do not strictly require the potential to be convex since it is guaranteed to be invertible in the infinitesimal setting of continuous normalizing flows (Chen et al., 2018). There exist works (Yang & Karniadakis, 2019; Finlay et al., 2020b; Onken et al., 2020) that have applied the theory of optimal transport to regularize continuous-time flows to have low transport cost. In contrast, we connect optimal transport with discrete-time normalizing flows, and CP-Flow is guaranteed by construction to converge pointwise to the optimal mapping between distributions without explicit regularization (see Section 4).

## 4    THEORETICAL ANALYSES

As explained in Section 2, the parameterization of CP-Flow is inspired by the Brenier potential. So naturally we would hope to show that (1) CP-Flows are distributionally universal, and that (2) the learned invertible map is optimal in the sense of the average squared distance the input travels $\mathbb{E}[||x - f(x)||^2]$. Proofs of statements made in this section can be found in Appendices C and D.

To show (1), our first step is to show that ICNNs can approximate arbitrary convex functions. However, convergence of potential functions does not generally imply convergence of the gradient fields. A classic example is the sequence $F_n = \sin(nx)/\sqrt{n}$ and the corresponding derivatives $f_n = \cos(nx)\sqrt{n}$: $F_n \to 0$ as $n \to \infty$ but $f_n$ does not. Fortunately, convexity allows us to control the variation of the gradient map (since the derivative of a convex function is monotone), so our second step of approximation holds.

**Theorem 2.** *Let $F_n : \mathbb{R}^d \to \mathbb{R}$ be differentiable convex functions and $G : \mathbb{R}^d \to \mathbb{R}$ be a proper convex function. Assume $F_n \to G$. Then for almost every $x \in \mathbb{R}^d$, $G$ is differentiable and $f_n(x) := \nabla F_n(x) \to \nabla G(x) =: g(x)$.*

Combining these two steps and Brenier's theorem, we show that CP-Flow with softplus-type activation function is distributionally universal.

**Theorem 3** (**Universality**). *Given random variables $X \sim \mu$ and $Y \sim \nu$, with $\mu$ being absolutely continuous w.r.t. the Lebesgue measure, there exists a sequence of ICNN $F_n$ with a softplus-type activation, such that $\nabla F_n \circ X \to Y$ in distribution.*

**N.B.**    In the theorem we do not require the second moment to be finite, as for arbitrary random variables we can apply the standard truncation technique and redistribute the probability mass so that the new random variables are almost surely bounded. For probability measures with finite second moments, we indeed use the gradient map of ICNN to approximate the optimal transport map corresponding to the Brenier potential. In the following theorem, we show that the optimal transport map is the only such mapping that we can approximate if we match the distributions.

**Theorem 4** (**Optimality**). *Let $G$ be the Brenier potential of $X \sim \mu$ and $Y \sim \nu$, and let $F_n$ be a convergent sequence of differentiable, convex potentials, such that $\nabla F_n \circ X \to Y$ in distribution. Then $\nabla F_n$ converges almost surely to $\nabla G$.*

The theorem states that in practice, even if we optimize according to some loss that traces the convergence in distribution, our model is still able to recover the optimal transport map, as if we were optimizing according to the transport cost. This allows us to estimate optimal transport maps without solving the constrained optimization in (2). See Seguy et al. (2018) for some potential applications of the optimal transport map, such as domain adaptation or domain translation.

## 5    EXPERIMENT

We use CP-Flow to perform density estimation (RHS of (1)) and variational inference (LHS of (1)) to assess its approximation capability, and the effectiveness of the proposed gradient estimator. All the details of experiments can be found in Appendix E. Code is available at https://github.com/CW-Huang/CP-Flow.

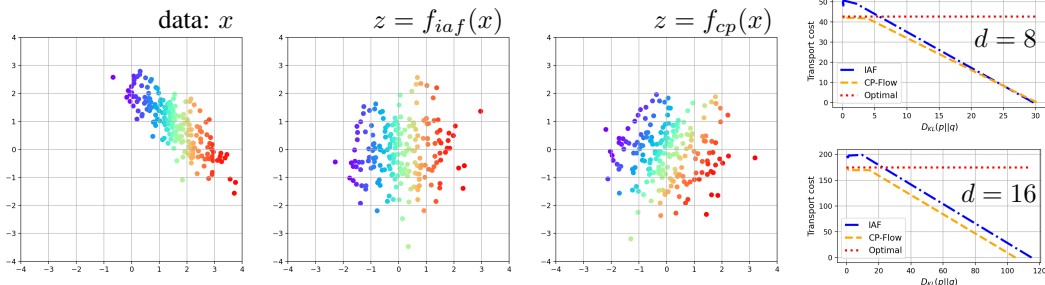

Figure 4: Approximating optimal transport map via maximum likelihood (minimizing KL divergence). In the first figure on the left we show the data in 2 dimensions. The datapoints are colored according to their horizontal values ($x_1$). The flows $f_{iaf}$ and $f_{cp}$ are trained to transform the data into a standard Gaussian prior. In the figures on the right, we plot the expected quadric transportation cost versus the KL divergence for different numbers of dimensionality. During training the KL is minimized, so the curves read from the right to the left.

**ICNN Architecture** Despite the universal property, having a poor parameterization can lead to difficulties in optimization and limit the effective expressivity of the model. We propose an architectural enhancement of ICNN, defined as follows (note the change in notation: instead of writing the pre-activations $z$, we use $h$ to denote the activated units):

$$F^{aug}(x) := L^+_{K+1}(h_K) + L_{K+1}(x)$$

$$h_k := \texttt{concat}([\widetilde{h}_k, h^{aug}_k]) \qquad \widetilde{h}_k := s(L^+_k(h_{k-1}) + L_k(x)) \qquad h^{aug}_k = s(L^{aug}_k(x)) \quad (7)$$

where half of the hidden units are directly connected to the input, so the gradient would have some form of skip connection. We call this the input-augmented ICNN. Unless otherwise stated, we use the input-augmented ICNN as the default architecture.

## 5.1 TOY EXAMPLES

Having distributional universality for a single flow layer means that we can achieve high expressiveness without composing too many flows. We demonstrate this by fitting the density on some toy examples taken from Papamakarios et al. (2017) and Behrmann et al. (2019). We compare with the masked autoregressive flow (MAF, Papamakarios et al. (2017)) and the neural autoregressive flow (NAF, (Huang et al., 2018)). Results are presented in fig. 3. We try to match the network size for each data. All models fit the first data well. As affine couplings cannot split probability mass, MAF fails to fit to the second and third datasets[2]. Although the last dataset is intrinsi-

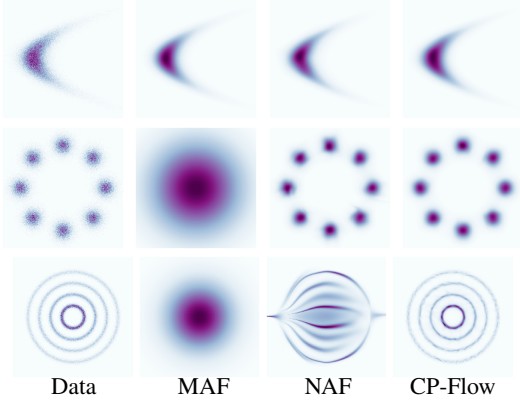

Figure 3: Learning toy densities.

cally harder to fit (as NAF, another universal density model, also fails to fit it well), the proposed method still manages to learn the correct density with high fidelity.

## 5.2 APPROXIMATING OPTIMAL COUPLING

As predicted by Theorem 4, CP-Flow is guaranteed to converge to the optimal coupling minimizing the expected quadratic cost. We empirically verify it by learning the Gaussian density and comparing the expected quadratic distance between the input and output of the flow against $J_{||x-y||^2}$ between the Gaussian data and the standard Gaussian prior (as there is a closed-form expression). In fig. 4, we see that the transport cost gets closer to the optimal value when the learned density

---

[2]Behrmann et al. (2019) demonstrates one can potentially improve the affine coupling models by composing many flow layers. But here we restrict the number of flow layers to be 3 or 5.

| Model | **POWER** | **GAS** | **HEPMASS** | **MINIBOONE** | **BSDS300** |
|---|---|---|---|---|---|
| Real NVP (Dinh et al., 2017) | -0.17 | -8.33 | 18.71 | 13.55 | -153.28 |
| FFJORD (Grathwohl et al., 2018) | -0.46 | -8.59 | 14.92 | 10.43 | -157.40 |
| MADE (Germain et al., 2015) | 3.08 | -3.56 | 20.98 | 15.59 | -148.85 |
| MAF (Papamakarios et al., 2017) | -0.24 | -10.08 | 17.70 | 11.75 | -155.69 |
| TAN (Oliva et al., 2018) | -0.48 | -11.19 | 15.12 | 11.01 | -157.03 |
| NAF (Huang et al., 2018) | -0.62 | -11.96 | 15.09 | 8.86 | -157.73 |
| CP-Flow (Ours) | -0.52 | -10.36 | 16.93 | 10.58 | -154.99 |

Table 1: Average test negative log-likelihood (in nats) of tabular datasets in Papamakarios et al. (2017) for density estimation models (lower is better). Standard deviation is presented in the appendix E.4.

| Model | **MNIST** | | **CIFAR-10** | |
|---|---|---|---|---|
| | Bits/dim | N. params | Bits/dim | N. params |
| Real NVP (Dinh et al., 2017) | 1.05 | N/A | 3.49 | N/A |
| Glow (Kingma & Dhariwal, 2018) | 1.06 | N/A | 3.35 | 44.0M[†] |
| RQ-NSF (Durkan et al., 2019) | — | — | 3.38 | 11.8M[†] |
| Residual Flow (Chen et al., 2019a) | 0.97 | 16.6M[‡] | 3.28 | 25.2M[‡] |
| Coupling Block Ablation | 1.02 | 3.1M | 3.58 | 2.9M |
| Residual Block Ablation | 1.04 | 2.9M | 3.46 | 3.1M |
| CP-Flow (Ours) | 1.02 | 2.9M | 3.40 | 1.9M |

Table 2: Negative log-likelihood (in bits) on held-out test data (lower is better). [†]Taken from Durkan et al. (2019). [‡]Obtained from official open source code.

approaches the data distribution (measured by the KL divergence). We compare against the linear inverse autoregressive flow (Kingma et al., 2016), which has the capacity to represent the multivariate Gaussian density, yet it does not learn the optimal coupling.

### 5.3  DENSITY ESTIMATION

We demonstrate the efficacy of our model and the proposed gradient estimator by performing density estimation on the standard benchmarks.

**Tabular Data**    We use the datasets preprocessed by Papamakarios et al. (2017). In table 1, we report average negative log-likelihood estimates evaluated on held-out test sets, for the best hyperparameters found via grid search. The search was focused on the number of flow blocks, the width and depth of the ICNN potentials. See appendix E.4 for details. Our models perform competitively against alternative approaches in the literature. We also perform an ablation on the CG error tolerance and ICNN architectures in appendix F.

**Image Data**    Next, we apply CP-Flow to model the density of standard image datasets, MNIST and CIFAR-10. For this, we use convolutional layers in place of fully connected layers. Prior works have had to use large architectures, with many flow blocks composed together, resulting in a large number of parameters to optimize. While we also compose multiple blocks of CP-Flows, we find that CP-Flow can perform relatively well with fewer number of parameters (table 2). Notably, we achieve comparable bits per dimension to Neural Spline Flows (Durkan et al., 2019)—another work promoting fewer parameters—while having using around 16% number of parameters.

As prior works use different architectures with widely varying hyperparameters, we perform a more careful ablation study using coupling (Dinh et al., 2014; 2017) and invertible residual blocks (Chen et al., 2019a). We replace each of our flow blocks with the corresponding baseline. We find that on CIFAR-10, the baseline flow models do not perform nearly as well as CP-Flow. We believe this may be because CP-Flows are universal with just one flow block, whereas coupling and invertible residual blocks are limited in expressivity or Lipschitz-constrained.

## 5.4 AMORTIZING ICNN FOR VARIATIONAL INFERENCE

Normalizing flows also allow us to employ a larger, more flexible family of distributions for variational inference (Rezende & Mohamed, 2015). We replicate the experiment conducted in Van Den Berg et al. (2018) to enhance the variational autoencoder (Kingma & Welling, 2013). For inference amortization, we use the partially input convex neural network from Amos et al. (2017), and use the output of the encoder as the additional input for conditioning. As table 3 shows, the performance of CP-Flow is close to the best reported in Van Den Berg et al. (2018) without changing the experiment setup. This shows that the convex potential parameterization along with the proposed gradient estimator can learn to perform accurate amortized inference. Also, we show that replacing the vanilla ICNN with the input-augmented ICNN leads to improvement of the likelihood estimates.

|                  | FREYFACES | OMNIGLOT | CALTECH |
|------------------|-----------|----------|---------|
| Gaussian         | 4.53      | 104.28   | 110.80  |
| Planar           | 4.40      | 102.65   | 109.66  |
| IAF              | 4.47      | 102.41   | 111.58  |
| Sylvester        | 4.45      | 99.00    | 104.62  |
| CP-Flow (vanilla) | 4.47     | 102.06   | 106.53  |
| CP-Flow (aug)    | 4.45      | 100.82   | 105.17  |

Table 3: Negative ELBO of VAE (lower is better). Standard deviation reported in appendix E.6.

## 6 CONCLUSION

We propose a new parameterization of normalizing flows using the gradient map of a convex potential. We make connections to the optimal transport theory to show that the proposed flow is a universal density model, and leverage tools from convex optimization to enable efficient training and model inversion. Experimentally, we show that the proposed method works reasonably well when evaluated on standard benchmarks.

Furthermore, we demonstrate that the performance can be improved by designing better ICNN architectures. We leave the exploration for a better ICNN and convolutional ICNN architecture to improve density estimation and generative modeling for future research.

## ACKNOWLEDGEMENTS

We would like to acknowledge the Python community (Van Rossum & Drake Jr, 1995; Oliphant, 2007) for developing the tools that enabled this work, including numpy (Oliphant, 2006; Van Der Walt et al., 2011; Walt et al., 2011; Harris et al., 2020), PyTorch (Paszke et al., 2019), Matplotlib (Hunter, 2007), seaborn (Waskom et al., 2018), pandas (McKinney, 2012), and SciPy (Jones et al., 2014).

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
