# OpenReview forum: "Convex Potential Flows: Universal Probability Distributions with Optimal Transport and Convex Optimization"
_ICLR.cc/2021/Conference — ICLR 2021 Poster_

### Official Review · AnonReviewer2 · 2020-10-25
**Official Blind Review**

**Rating:** 6
**Confidence:** 4

**Review:**

Summary:
This paper proposes the flow based representation of a probability distribution so that the corresponding density remains tractable. In particular, the push-forward map that generates the desired distribution is characterized by the gradient of a strongly convex potential function. The invertability of the mapping as well as the Jacobian of the mapping is hence guaranteed by such a convexity property of the potential function. The proposed CP-flows are proved to be universal density approximators and are optimal in the OT (2-Wasserstein) sense.

Comment:
It seems to me that the proposed CP-flow model is quite similar to a set of recent works that use the ICNN to parameterized the convex Brenier potential of 2-Wasserstein. This limits the novelty of this paper.
On the other hand, I think one contribution of this paper is to study how the gradient of the log determinant of the Hessian of the potential function can be computed, which relies on a standard technique, the Hutchinson trace estimator.
Another contribution of this paper is to study universality of the CP-flow model as a density approximators. Since any convex function yields an affine-max representation which is a special case of the ICNN, the Brenier's theorem implies the universality.

Concerns:
1. Unlike the traditional flow models which consists of a sequence of simple transformations, it seems to me that CP-flow is not exactly a "flow" model as it only involves a single (but complex) transformation (which of course contains multiple layers). Correspondingly, the computation involved in the flow model (inversion, gradient evaluation) are substantially more expensive.
2. The presentation of the paper focuses on the description of the model. However, I would suggest the authors to add a few paragraphs to briefly discuss some problems that takes the CP-flow model as a subroutine, e.g. variational inference or maximum likelihood loss. This would make the paper more self-contained and it will also justify the need to have a traceable density.
3. More on the writing of the paper. Some derivations to show that why (3) is important are appreciated, e.g.
$det \frac{\partial f^{-1}(x)}{\partial x} = \frac{1}{det(\nabla f(f^{-1}(x)))}$. Again, it would be good for a broader audience if the paper is more self-contained.
4. How to efficiently implement the batch computation of the gradient of the model? What do you mean by "Inverting a batch of independent inputs is as simple as summing the convex potential over all inputs"?

*********
The authors partially addressed my concerns. Therefore I raise my score to 6.

---

> ### Author Response · Authors · 2020-11-16
> **Official response to R2**
>
> We thank the reviewer for the feedback and we’d like to address the concerns about novelty and some other questions the reviewer has below.
>
> > It seems to me that the proposed CP-flow model is quite similar to a set of recent works that use the ICNN to parameterized the convex Brenier potential of 2-Wasserstein. This limits the novelty of this paper.
>
> The novelty is not merely to parameterize an optimal transport map, but also to realize that it can be made invertible and trained with maximum likelihood or KL as a normalizing flow. Our primary intention is to bring novelty from the OT theory into the flow community to formulate a natural design of invertible models that has theoretical guarantees. Moreover, we’ve derived a new gradient estimator that also leverages tools from the optimization literature, which makes likelihood-based learning computationally tractable. We emphasize that being able to approximate the optimal transport map (with only one flow layer) is only an interesting and potentially useful property that CP-Flow possesses, which is not to be confused with the key contribution of the work.
>
> > CP-flow is not exactly a "flow" model as it only involves a single (but complex) transformation
>
> We actually compose multiple CP-Flow layers (like a flow), wherein the surrogate of the log-determinant gradient estimate is computed separately (each calling Algo 2 once) and summed up as the total surrogate for the logdet.
>
> > computation involved in the flow model (inversion, gradient evaluation) are substantially more expensive
>
> Having an inner computation routine is not new in the normalizing flow literature. For example, Residual Flow also has an inner routine that estimates the vector-Jacobian inverse product using the Roussian Roulette (RR) estimator. The benefit here is that the same problem can be solved via convex optimization (most specifically conjugate gradient) with fast convergence guarantee (the estimation time is bounded, as opposed to the stochastic truncation done by RR).
>
> CP-Flow is also not the only flow-based method that requires an iterative procedure for inversion; examples include Residual Flow (aka i-ResNet, using fixed point), Implicit normalizing flow (using fixed point), neural autoregressive flows (with bijection search, or cast as the derivative of a convex potential as discussed in our paper) or any autoregressive flow that needs O(d) for inversion, MintNet (using fixed point with a local but not global convergence guarantee), generalized sylvester with matrix/convolution exponential (using fixed point). The benefit of CP-Flow is, again, that it uses a black-box convex solver which can solve for the inversion efficiently.
>
> > add a few paragraphs to briefly discuss some problems that takes the CP-flow model as a subroutine, e.g. variational inference or maximum likelihood loss
>
> We now allude to equation (1) in the experiment section to emphasize the gradient of these two identities are approximated during training. We would love to update the paper more should you have more suggestions to help refine the readability of the paper.
>
> > Some derivations to show that why (3) is important are appreciated
>
> The importance is that as long as we can unbiasedly estimate or compute $v^\top H^{-1}$ exactly, we would have an “unbiased” and “constant-memory” estimator of the gradient. The constant-memory property is not emphasized in the main text, we’ll include it. Thanks for the suggestion. This is in contrast to simply backpropagating through an estimator of the logdet, which has ill-defined derivatives and requires significantly more memory (shown in Figure 2).
>
> > det∂f−1(x)∂x=1det(∇f(f−1(x)))
>
> We are not sure where you think this identity is used in the derivation of (3).
>
> > How to efficiently implement the batch computation of the gradient of the model? What do you mean by "Inverting a batch of independent inputs is as simple as summing the convex potential over all inputs"?
>
> Inversion: Since all of the entries of the scalar l in the minibatch are independent of each other, computing the gradient all l's wrt all x's amounts to computing the gradient of the summation of l's wrt all x's.
>
> Gradient estimation: Similar to inversion, the "hvp" in the surrogate loss can also be computed in batch by summing over the data index, since all entries are independent. This is summarized in the following snippet (using PyTorch), which is how we implement the batch hvp in operation. The grad_output in PyTorch has the same effect as taking the dot product with v and summing over the entire batch, followed by differentiation.
>
> ```
> def hvp_fun(v):
>     # v is the rademacher RV
>     v = v.reshape(batch_size, *dims)
>     # f is the gradient map, x is an input that requires gradient
>     hvp = torch.autograd.grad(f, x, grad_output=v, create_graph=True, retain_graph=True)[0]
>     return hvp.reshape(batch_size, -1)
> ```
>
> We have updated the paper to clarify this point; thank you for the question.

---

> > ### Comment · AnonReviewer2 · 2020-11-18
> > **response to the authors**
> >
> > >We actually compose multiple CP-Flow layers (like a flow), wherein the surrogate of the log-determinant gradient estimate is computed separately (each calling Algo 2 once) and summed up as the total surrogate for the logdet.
> >
> > Since the family of the gradients of convex functions is not closed under composition, it seems that the theoretical analysis does not cover the case when multiple CP-Flow layers are composed.

---

> > > ### Author Response · Authors · 2020-11-18
> > > **Clarification on universality (not using composition)**
> > >
> > > Indeed, the gradients of convex potentials are not closed under composition, but this only says we can approximate a potentially larger family of functions through composition, and the theory is still correct in saying the flow is universal.
> > >
> > > To see the later, we can let all the convex potentials beyond the first flow layer be $\frac{1}{2}||x||^2$, which simply gives us an identity map. That is, as long as a single flow (in this example, the first flow layer) is universal, so is a chain of flows.
> > >
> > >
> > > This is similar to the analysis of universal result of *neural autoregressive flows* and *sum-of-squares polynomial flows*. They show that single autoregressive flows (without composition) can approximate arbitrary distributions. At the same time, composing autoregressive flows with index reversing / permutation will render the overall transformation no longer triangular. But composition only improves expressivity since all these flows can represent an identity map.

---

### Official Review · AnonReviewer1 · 2020-10-25
**Official Blind Review #1**

**Rating:** 7
**Confidence:** 4

**Review:**

#### Summary and contributions
This paper presents CP-Flow, an alternative formulation of normalizing flows via the gradient of a strictly convex function. Concretely, an optimal transport map is constructed for the quadratic transport cost. This map can be constructed via the gradient of a strictly convex function owing to Brenier’s Theorem. The strict convexity of the potential function ensures that the gradient map is injective and surjective (i.e., invertible). This leads to relations with normalizing flows, invertible maps that can be trained via exact maximum likelihood. The authors borrow techniques from convex optimization to invert the gradient map and to compute the log-determinant of the Hessian (and its gradient), quantities that are required for density evaluation and gradient-based training. Using arguments from optimal transport literature, the authors prove the universality and optimality of the gradient maps when they are realized using input-convex neural networks (ICNNs). Experiments in various settings of density estimation and variational inference have been presented that demonstrate a competitive performance of CP-Flow against competing normalizing flow models.

#### Strengths
This work makes a significant contribution to the body of literature on optimal transport-based generative models. By parameterizing the flow as the gradient of a convex potential function ($R^d \rightarrow R$), this method improves the parameter efficiency of invertible models. Moreover, the network architectures, although restricted to ICNNs, can be much more flexible than in conventional normalizing flows where networks have to satisfy restrictive conditions to ensure invertibility. Theoretical results also demonstrate that ICNNs have the capacity to learn transport maps between arbitrary continuous probability measures. The theoretical results are supported by the empirical evaluation which, despite lacking scale, is sufficient to demonstrate the potential of the proposed method.

#### Weaknesses
The primary weakness of this work is that some directly related works that utilize Brenier’s Theorem to learn transport maps via ICNNs have not been discussed in sufficient detail. Apart from flow-based setups, other works have also explored the construction of transport maps using ICNNs which have not been discussed (see for example [1, 2, 3]). Moreover, the experiments lack comparisons with these works or a discussion of why such comparisons are not possible.  Another recent related work is [4].
A potential concern is the efficiency of solving a quadratic optimization problem for computation of the gradient of the log-det of Hessian for each gradient step. Did the authors compare the runtimes of the different methods?

[1] Korotin, Alexander, et al. "Wasserstein-2 Generative Networks." arXiv preprint arXiv:1909.13082 (2019).
[2] Taghvaei, Amirhossein, and Amin Jalali. "2-wasserstein approximation via restricted convex potentials with application to improved training for gans." arXiv preprint arXiv:1902.07197 (2019).
[3] Makkuva, Ashok Vardhan, et al. "Optimal transport mapping via input convex neural networks." arXiv preprint arXiv:1908.10962 (2019).
[4] Finlay, Chris, et al. "Learning normalizing flows from Entropy-Kantorovich potentials." arXiv preprint arXiv:2006.06033 (2020).


#### Additional feedback

Suggestions:
- Include a proper discussion of related works that use ICNNs to parameterize the Brenier potential and compare with these methods experimentally, if possible.

Questions:
- Is there a reason that the authors did not include a NSF ablation in Table 2?
- Did the authors investigate the quality of generated samples for the image datasets (using FID, IS, etc.) as the poor quality of images is a known problem for flow-based models?
-------
The authors have addressed most of my concerns. I believe the paper is a good contribution to the literature on normalizing flows; therefore, I firmly vote for acceptance.

---

> ### Author Response · Authors · 2020-11-15
> **Official response to R1**
>
> #### Related work using ICNNs
>
> We’d like to thank the reviewer for the positive feedback. We’ve added the references and discuss the difference in the updated version of the paper (we added the first three to section 2 where we talk about OT, and the last one to 3.3 under Flows with Potential Parameterization). The reason why a comparison with those papers is not provided is because the cited works are based on approximating the 2-Wasserstein distance directly for generative modelling, whereas we train our methods using KL divergence via MLE. We believe a comparison with some other flow-based method trained to maximize the likelihood of the data is more relevant, as our work is about the design of a new family of flows. Nonetheless, our methodology can be still viewed as a contribution to OT map approximation for high-dimensional data, as the reviewer has noted.
>
> #### Runtime of gradient estimate (CG iterates)
>
> We’ve analyzed the # of CG iterates as a function of training time for different absolute tolerance specifications in Figure 8 in the appendix. We’ve also included two new sub-figures to study the increase in runtime for smaller tolerance values in the updated version.
>
> Theoretically, we can plug in any convex solver to solve for the minimizer of the quadratic problem; e.g. we can also use the l-BFGS algorithm just like for inversion. We choose the conjugate gradient method because CG is guaranteed to converge within d steps, modulo the error of numerical rounding (where d is dimensionality).
>
> #### NSF ablation
>
> We did not directly compare with NSF since it uses a more sophisticated transformer as well as NN block (the NN block used in NSF is different from the original Glow’s implementation).
>
> #### Sample quality
>
> We did not include the generated samples in the manuscript as we do not find them intriguing. We believe the sample quality of CP-Flow can be further improved by designing better convolutional ICNN architecture for modeling imagery data. We hope our work motivates the search for better input-convex architectures in the future.

---

### Official Review · AnonReviewer4 · 2020-10-27

**Rating:** 5
**Confidence:** 4

**Review:**

### Summary
The authors introduce CP-Flows, a way to parameterize normalizing flows by constructing an input-convex neural net with softplus-type activation functions and considering its gradient as the flow. They add a quadratic term to ensure invertibility. Using convex optimization techniques their method only needs access to convex optimization solvers. They show that this architecture is universal (that is, starting from a measure $\mu$, there is a sequence of CP-Flows converging weakly to a desired distribution $\nu$). They also prove that the constructed flow converges pointwise to the optimal Brenier map for Euclidean cost. They perform a set of experiments on synthetic and real-world datasets, and show their method delivers its promises.

### Strong/Weak Points
+ The ideas described in the paper are simple and easy to understand.
+ The paper is generally well-written (with exceptions detailed below).
+ The ideas for computing the trace of log Hessian and its gradient are neat; however, I am not sure if they are novel and not present in the literature.
+ All the results are asymptotic. The convergence results are weak: convergence to the optimal map (Theorem 4) is point-wise, and convergence of the distributions is weak (Theorem 3).
+ There are lots of tricks used here and there (computing the trace of log Hessian and its gradient) to reduce computational complexity, but nothing explicit is computed theoretically nor presented experimentally.
+ I am not sure if Theorem 2 and Theorem 4 are already known or not. I might not recall the reference, but I am pretty confident that they have been existing in the literature.
+ In the experiments, looking at Table 1, NAF always outperfrom CP-Flow (and no explanation has given for why is it the case). The same is for Table 3, where Sylvester outperforms CP-Flow in every dataset.

I have decided to give a 5 to this paper, as theoretically it does not add significantly to the current theory of OT and NFs. Experimentally, I feel that the performance is (probably) marginally better than the other methods if it is not worse.

### Additional Feedback
Fixing the following errors improves readability of the paper:
- Section 1, paragraph 3, line 4, "the network" has not been introduced and not clear where it is referring to.
- same, line 7, universality and optimality is not clear (in what sense). It becomes clear later.
- page 2, line 2, "the network architecture" still is dangling.
- Section 2, two lines above "Universal Flows" does not make sense. "ordinary differential equations, orthogonality or Lipschitz constants" are not NN architectures.
- same, two lines above "Optimal Transport" there is no reference why "under very mild conditions a coupling exists".
- same, "Optimal Transport". The problem described is not Monge's problem. It's Monge's formulation for optimal transportation.
- page 3, "Invertibility and ..." the activation changed from $s$ to $g$ suddenly.
- page 4, lines 11, 12 are hard to understand. Why NaNs creep in?
- page 4, (3), why the terms are in red?
- page 5, "Sigmoidal Flow" activation is called $s$ while in the formula it is $\sigma$.
- same, "Flows with Potential ..." the last two lines, I do not see what the contrast is made against.
- page 6, Section 5, I do not see why the notation changes from $z$ to $h$.
- page 7, I do not know how to interpret the middle plots for $f_\mathrm{iaf}(x)$ and $f_\mathrm{cp}(x)$.

---

> ### Author Response · Authors · 2020-11-17
> **Official response to R4**
>
>
> > theoretically it does not add significantly to the current theory of OT and NFs
>
> **NFs:** Thank you for your feedback and criticism. This indeed is not meant to be a theory paper. However, we do believe we have made strong contributions to the literature on normalizing flows and likelihood-based modeling. Our work is meant to bring theory from optimal transport and convex optimization techniques into the normalizing flow community, so as to design a flow in an arguably natural way that has theoretical benefits: universality and optimality. To the best of our knowledge, we’re the first to use the gradient of a convex potential to parameterize a normalizing flow and at the same time propose a gradient estimator which can be tractably computed to enable likelihood-based training.
>
> **OT:** Our work is the first likelihood-based generative model for high-dimensional data which provably delivers OT maps upon convergence. Recent work on 2-Wasserstein models [1-3] has attempted to approximate the 2-Wasserstein distance using ICNN, and by approximating the distance they obtain the optimal transport map by taking the gradient. On the contrary, our work obtains pointwise convergence towards the optimal transport map upon distributional convergence, which in our case is using the (empirical) KL divergence.
>
> [1] Wasserstein-2 Generative Networks
>
> [2] Optimal transport mapping via input convex neural networks
>
> [3] 2-Wasserstein Approximation via Restricted Convex Potentials with Application to Improved Training for GANs
>
> > but nothing explicit is computed theoretically nor presented experimentally.
>
> Theoretically, the gradient estimator takes constant memory and takes O(m) time to compute, where m is the number of CG iterates, which depends on the user-defined tolerance value so it terminates adaptively. We did include an empirical ablation study on tuning the absolute tolerance value in appendix E; see Fig 8 for an analysis on # CG iterates vs tolerance. We also added new subfigures showing the effect on per-iteration time.
>
> > I am not sure if Theorem 2 and Theorem 4 are already known or not. I might not recall the reference, but I am pretty confident that they have been existing in the literature.
>
> We have tried our best trying to find relevant references to cite. The proof of theorem 2 heavily relies on the monotonicity of the (sub-)derivative of convex functions, and some form of it can probably be found in exercises of standard analysis textbooks with different conditions using similar proof techniques. The proof of theorem 4 relies on a converse result of Brenier’s theorem which we cited in our proof. If you could refer to additional references that include a statement closer to ours, we would love to cite them.
>
> > In the experiments, looking at Table 1, NAF always outperfrom CP-Flow (and no explanation has given for why is it the case). The same is for Table 3, where Sylvester outperforms CP-Flow in every dataset.
>
> The density estimation experiment (Table 1) is a well-established benchmark for flow-based models, and some architectures including autoregressive models have been well explored by a lot of prior work. This partly explains why NAF (which relies on an autoregressive conditioner) outperforms CP-Flow on that task. While we intend to show that CP-Flow can approximate the density well with the proposed gradient estimator, we do not intend to beat the SOTA with an architecture like ICNN that is still relatively under-explored (compared to autoregressive models).
>
> The goal of the VAE experiment (Table 3) is to show that CP-Flow can be used as an accurate black-box inference machine off the shelf. While it needs not be the case, the likelihood estimate of VAE tends to be improved with more accurate approximate posterior. The exact performance of VAE might still be subject to other factors including KL annealing (which introduces bias), amortization architecture, etc. We did not further tune these other factors.
>
> > page 7, I do not know how to interpret the middle plots for fiaf(x) and fcp(x).
>
> Figure 4 is meant to demonstrate how different families of normalizing flows can learn to transform a distribution (Figure 4-left) into another (standard normal, the 2nd and 3rd), while they can be different functions. It shows that an autoregressive flow (iaf) changes the horizontal coordinate (x1) marginally, and redistributes the vertical coordinate (x2) conditionally on the x1. Therefore if we color the data according to the value of x1, after the transformation all data points having the same z1 originally have the same x1 and thus have the same color. On the other hand CP-Flow learns the optimal transport map that minimizes the quadratic cost, and does not preserve the color shared by the horizontal coordinates. This is to demonstrate qualitatively how two universal flows learn to approximate different transport maps between probability distributions, and that CP-Flow learns the optimal map.

---

> ### Author Response · Authors · 2020-11-17
> **Official response to R4 (more clarification on additional feedback)**
>
> **Additional feedback on editing:** We have updated the paper taking into account your suggestions.
>
> Some clarifications
>
> > page 4, lines 11, 12 are hard to understand. Why NaNs creep in?
>
> Gradients through eigendecomposition are ill defined when there are degenerate eigenvalues (see Seeger et al., 2017). PyTorch handles these cases by simply returning a nan gradient.
>
> > page 4, (3), why the terms are in red?
>
> We colored some parts of (3) in red to make it easier to identify the adjugate representation of matrix inverse.
>
> > "Flows with Potential ..." the last two lines, I do not see what the contrast is made against.
>
> This is in contrast to the cited works just one sentence above, which regularize the flow to be optimal, whereas (one layer of) CP-Flow is always an optimal map between the input measure and its pushforward.
>
> > page 6, Section 5, I do not see why the notation changes from z to h
>
> It’s simply because it’s easier to write a recursion formula this way (with the augmented variables). It’s a notational convenience.

---

> ### Author Response · Authors · 2020-11-23
> **Official response to R4 (novelty of the gradient estimator)**
>
> > The ideas for computing the trace of log Hessian and its gradient are neat; however, I am not sure if they are novel and not present in the literature.
>
> We'd like to emphasize again that one of our contributions is to use a convex optimization algorithm as subroutine to compute stochastic gradients required for likelihood optimization.
>
> The problem we addressed is that (1) we circumvent backpropagating through a numerically unstable, potentially biased estimator using the stochastic Lanczos quadrature method [1], and (2) and replace it with a direct gradient estimator that is constant-memory (compared to Lanczos, which can be shown to take $\mathcal{O}(m^2)$ memory) and numerically unbiased.
>
> The key novelty is that we take advantage of the structure of the Jacobian (which is the Hessian) being symmetric positive-definite, and apply the conjugate gradient method to solve the vector-inverse-Hessian product. This is in contrast with the prior work (Residual Flow, [2]) that also uses a similar formula, but has to use a stochastic estimator (Russian roulette estimator) to unbiasedly estimate the said quantity.
>
> [1] Fast estimation of tr(f(A)) via Stochastic Lanczos Quadrature
>
> [2] Residual flows invertible generative modeling

---

### Official Review · AnonReviewer3 · 2020-11-05
**gradient map of a convex potential function as a flow**

**Rating:** 8
**Confidence:** 4

**Review:**

#####################

Summary and contributions:

The paper provides a novel approach to normalizing flows. It models a normalizing flow as the gradient map of a convex potential function.  The gradient estimation and model inversion, that are the computationally expensive part of arbitrary complex NFs, are formulated as cheap convex optimization problems. It proves that  the proposed CP-flow is a universal density approximator and also shows that it is an optimal transport (OT) map.

#####################

Reasons for score:

I vote for clear acceptance. The idea of convex potential flow sounds very interesting and the way the computational complexities are very useful for designing more advanced NFs. The paper is very well written.

Strength:

1. The paper is very well written and structured and is easy to read for a wide audience. It also provides a good review of main papers in the literature
2. The motivation for using CP-flow is well explained.
3. The claims are well supported by theoretical proofs and empirical studies.
4. The gradient estimation and model inversion, that are the computationally complex part of arbitrary complex NFs, are formulated as convex optimization problems that gain advantage of fast converging and cheap optimization algorithms and it also leverages the efficient Hessian-vector product computation.
5. The experimental results show the proposed flow can perform competitively with much less number of parameters .

#####################

Additional Feedback and Questions:

1. After reading the paper it is not quite clear why we do need an optimality in Monge sense (Theorem 4) and what is the point of optimal transport in this work. It is worth expanding or adding more insights to the motivation given in the Introduction by the notion of rearrangement cost.
2. Orthogonality or Lipschitzness constraints are mentioned without citation in introduction.
3. What is so special about the soft-plus type nonlinearity in theorem 3, can we use other non-linearities such as the symmetric-log derived in [1] that are monotonic and differentiable by construction ?
4. CP-flow looks more expressive than NAF in toy examples but why is it outperformed in density estimation? As a more insightful comparison, I suggest comparing the “transport cost” of the NAF with that of ICNN in section 5.2 (Figure 4) as NAF is outperforming ICNN in all the benchmarks and is universal.
5. It is worth comparing the number of parameters of the optimal CP-flow in Table 1 with the available methods to have a better understanding of models’ flexibility. Maybe, CP-flow can achieve the SOTA if the number of parameters are normalized.
6. I wonder if CP-flow can outperform the Residual flow which uses a similar gradient estimator routine if its number of parameters, in Table 2, is increased.
7. Compared to RealNVP, CP-flow requires an optimization solution per each training update. Also, as noted in the paper, the computational cost of the CP-flow is less than residual flows as it saves the Spectral normalization process. So how the speed and convergence rate of CP-flow is compared against the benchmarks, assuming all are using the same hardware e.g GPUs (and cvx opt are implemented in GPU). It looks better to have a sense of it in one of the experiments.
8. I am willing to see the randomly generated sample of the CP-flow especially to compare its local/global stability against the Residual flow due to its Lipschitz constraint as discussed in [2].
9. A schematic architecture of the ICNN model helps better understand it.

Ref:
[1] M. Karami, D. Schuurmans, Jascha Sohl-Dickstein, Daniel Duckworth, Laurent Dinh, “Invertible Convolutional Flow", Advances in Neural Information Processing Systems (NeurIPS) 2019,

[2] Behrmann, Jens, et al. "Understanding and mitigating exploding inverses in invertible neural networks." arXiv preprint arXiv:2006.09347 (2020)

---

> ### Author Response · Authors · 2020-11-18
> **Official response to R3**
>
> Thank you for your positive and detailed feedback. We are glad that you like our paper.
>
> > After reading the paper it is not quite clear why we do need an optimality in Monge sense (Theorem 4) and what is the point of optimal transport in this work.
>
> We indeed did not motivate the parameterization of CP-Flow using optimality, but instead we use it as a universal transport map and approximate it using the gradient of a convex potential to obtain distributional universality. Optimal transport maps have found some use cases in machine learning though, as they can be used to perform unsupervised domain adaptation [1-3] and domain translation [4]. We’ll add some reference in the theoretical section for the interested readers.
>
> [1] Optimal Transport for Domain Adaptation
>
> [2] Mapping Estimation for Discrete Optimal Transport
>
> [3] Large Scale Optimal Transport and Mapping Estimation
>
> [4] Optimal Unsupervised Domain Translation
>
>
> > What is so special about the soft-plus type nonlinearity in theorem 3, can we use other non-linearities such as the symmetric-log derived in [1] that are monotonic and differentiable by construction ?
>
> We use softplus type activation because 1) it is convex and monotone, which is required if we want ICNN to be convex, and 2) softplus type activations can approximate ReLU arbitrarily well via rescaling, and this is part of the proof of universality in Prop 3. We showed that we can generate a large sub-family of softplus type activation functions by taking the antiderivative of CDFs in Prop 1. This tells us this family of activation functions is quite expressive, as one can transform an arbitrary 1D distribution into a uniform distribution via its CDF.
>
> S-Log is not convex, so it cannot be used as the activation function of ICNN. One can potentially use non-softplus type activation that is still monotone and convex, but we did not explore much in this direction.
>
> > CP-flow looks more expressive than NAF in toy examples but why is it outperformed in density estimation? … comparing the “transport cost” of the NAF with that of ICNN in section 5.2 (Figure 4) as NAF is outperforming ICNN in all the benchmarks and is universal.
>
> Both are universal density models. But we find the last toy example particularly hard to approximate using NAF, perhaps because the “conditional transformation” it needs to approximate is more complex.
>
> We believe the better performance of NAF on density estimation is partly due to the use of an autoregressive conditioner, which is a well-explored neural network architecture family, besides its capability to model multimodal data using non-linear transformers. CP-Flow relies on the ICNN architecture, which is a relatively underexplored topic. We hope to motivate the design of more powerful ICNN architectures so that it can be on par with SOTA density models.
>
> As of Fig 4, the best NAF can do in this case is actually learning a linear transformer, since both the input & target distributions are Gaussian, which is why we believe a linear autoregressive flow is sufficient.
>
> > It is worth comparing the number of parameters of the optimal CP-flow in Table 1 with the available methods to have a better understanding of models’ flexibility.
>
> When choosing the hyperparameter grid, we make it so that the sizes of the networks of the grid more or less match the sizes of the networks of the grid of the MAF paper. So we did not compare the parameter counts.
>
> > I wonder if CP-flow can outperform the Residual flow which uses a similar gradient estimator routine if its number of parameters, in Table 2, is increased.
>
> Note that while CP-Flow and Residual Flow use a similar gradient estimator, the two models are still significantly different from each other. In CP-Flow, our potential parameterization means we only require a neural network of R^d -> R, which typically has much fewer parameters than directly parameterizing the gradient map. We never trained a CP-Flow with as many parameters as the Residual Flow baseline, but the emergence of stronger ICNN architectures might make this comparison worthwhile.

---

> ### Author Response · Authors · 2020-11-18
> **Official response to R3 (continued)**
>
> > Compared to RealNVP, CP-flow requires an optimization solution per each training update... So how the speed and convergence rate of CP-flow is compared against the benchmarks...
>
> Training of RealNVP is much faster, and CP-Flow and residual flow have similar speed during training. This is because residual flow uses the russian roulette estimator and one can control the average truncation index to be small to save compute time. We can potentially apply the same trick to truncate the CG iterates, trading off variance for compute time, and making CP-Flow faster than residual flow, but we do not find this necessary in our experiment and did not pursue this direction. On the other hand, the training of CP-Flow can also be sped up by letting CG terminate earlier (which introduces bias). For an analysis on the speed of CP-Flow with different tolerance values, which control the adaptive termination of the algorithm, please see Fig 8 in the appendix.
>
> > stability against the Residual flow
>
> CP-Flow can faithfully reconstruct the in-distribution samples, as shown by the reconstructions in Fig 6. We also added a new analysis on the reconstruction loss (Fig 2 of Behrmann 2020) of different input distributions; the result is summarized in Table 8. CP-Flow can reconstruct to high precision by setting a smaller error tolerance value for the convex optimizer and does not have the exploding behavior of coupling blocks of Glow/RealNVP. Thank you for your question.
>
> > schematic architecture of the ICNN model
>
> A visualization of the vanilla ICNN architecture can be found in Fig 1 and 2 of [5].
>
> [5] Input Convex Neural Networks https://arxiv.org/pdf/1609.07152.pdf

---

### Public Comment · ~Cheng_Lu5 · 2020-11-11
**Very interesting work! Some questions about the proof of university**

The idea for constructing a NF by strongly convex function is very interesting, and the use of OT brings a new perspective for the capacity for NFs. However, I have a question about the proof of university. The invertibility in Sec.3.1 seems to be a conflict with the universality in Sec.4

For CP-Flow, we need to add a quadratic term $F_\alpha(x)=\frac{\alpha}{2}\|x\|_2^2+F(x)$ to ensure that the convex function is strongly convex and then the gradient is invertible. This is necessary for the invertibility of CP-Flow. However, to prove the universality, you construct a convex piecewise linear function under the formulation of CP-Flow. But this function sometimes is not strongly convex, so the invertibility of such a convex piecewise linear function is not ensured.

The key point is, in Thm2 you want to prove the gradients of a family of strongly convex functions can approximate any gradient of all convex functions. However, the gap between strongly convex function and convex function cannot be ignored. Let's take a simple example for $d=1$. Let $G(x)=e^{-x}$, then $\nabla G(x)=-e^{-x}$ and the Hessian is $H(x)=e^{-x}$. However, $\lim_{x\rightarrow \infty}H(x)=0$, so $G(x)$ is not strongly convex. For any CP-Flow $F(x)$ with $\alpha$ w.r.t. the quadratic term, the gap between $\nabla F(x)$ and $\nabla G(x)$ is bounded by a constant of $\alpha$. So the university of CP-Flow is not ensured.

I wonder if this problem is a misunderstanding of your nice work. And I really appreciate it if you could give me some more explanations. Thank you!

---

> ### Author Response · Authors · 2020-11-17
> **Good question!**
>
> Hi Cheng Lu, thank you for your interest in our paper and your question! The example you give might help other readers understand the concept of approximation better when reading our paper.
>
> First of all, the $\alpha$ in our paper is learnable and is not fixed. Having a non-zero $\alpha$ is just to ensure the gradient map is indeed invertible, no matter how small it is. This can be done by using a non-linear activation function to reparameterize a trainable scalar parameter, e.g. softplus.
>
> Second, since $\alpha$ now is not fixed, we can take it to be arbitrarily close to zero, so that we can still have a sequence of "invertible" maps that converge pointwise to the desired transport map. We can do so by tweaking the proof a bit as follows. Assuming $f_n$ is the sequence that we take to converge to a transport map $g$ which pushes some input measure onto some desired output measure. Let $F_n$ be the convex potential of $f_n$. Then we can set $F'_n= \frac{1}{2n}||x||^2 + F_n$, and thus $f'_n = \frac{1}{n} x + f_n$. This way, $F'_n$'s are strongly convex, and $f'_n$'s are invertible. And since $\frac{1}{n} x\rightarrow 0$, we have $f'_n \rightarrow 0 + g = g$.
>
>
> Using your example, we can let $f'_n = \frac{1}{n} x  - e^{-x}$ (assuming $F_n=G$ for simplicity). The image of $f'_n$ is the entire $\mathbb{R}$ and $f'_n$ is invertible while $g=-e^{-x}$ maps $\mathbb{R}$ to $(-\infty, 0)$ and is not invertible (it's not surjective). But for any $x$, by taking $n\rightarrow\infty$ we still have
>
> $$\lim_{n\rightarrow\infty} f'_n(x) = g(x)$$
>
> That is, for the purpose of pointwise approximation (pointwise convergence), we can take a restricted subset of functions (e.g. strongly convex functions) to approximate a larger family of functions (e.g. convex functions).

---

### Author Response · Authors · 2020-11-24
**Summary of changes**

We would like to thank all the reviewers for the feedback and suggestions. We've taken them into account when updating the paper.

1) We'd like to reiterate the main contribution and novelty of this work is to motivate a new parameterization of flow-based model using optimal transport theory, and we make use of convex optimization techniques to enable gradient estimation for likelihood maximization. Theoretically, the proposed flow is universal and optimal in the OT sense. Practically, the flow can be trained efficiently using CG to estimate the gradient of the log-determinant of Jacobian, which is typically very expensive (or even too big to fit in memory!).

We added a few more plots to study the runtime of optimizing the flow using the proposed gradient estimator (with different atol specifications) and a baseline estimator using Lanczos (with different eigenvalues) which is problematic as described in section 3.2. More specifically,

2) We added new subfigures (of Figure 8) in appendix E to analyze the effect of tuning atol value for CG on run-time. It shows the runtime increases with higher precision requirement, and saturates when m reaches the dimensionality of the data.

3) Furthermore, we compare the above against "directly back-propagating through the Lanczos estimator" mentioned at the end of section 3.1 and beginning of 3.2. We look at the effect of the number of eigenvalues used in the stochastic Lanczos quadrature  (SLQ) method (i.e. size of the tridiagonal matrix) on training time. In this experiment we directly back-propagated through the Lanczos estimate to compute the stochastic gradient (instead of using the gradient estimator proposed in Section 3.2). Figure 9 shows that the runtime is much higher than the runtime of the proposed gradient estimator using CG (bottom left of Figure 8). We note that the experiments with m = 5 diverged, possibly due to the error in estimation. This complements the memory profile shown in Figure 2, and accentuates the lower runtime and memory requirement of the proposed method by contrast.

---

### Decision · Program_Chairs · 2021-01-07
**Final Decision**

**Decision:**

Accept (Poster)

**Comment:**

The main contribution of the paper is a novel parametrization of normalizing flows using ideas from optimal transport theory. This new parametrization allows viewing a normalizing flow as the gradient of a convex function, which leads to an efficient method for gradient estimation for likelihood estimation, using only access to convex optimization solvers. The paper is overall well-written and provides a clean analysis. Theoretical results from the paper are supported by experiments. The paper was overall viewed favorably by the reviewers.